# Effectors of *Puccinia striiformis* f. sp. *tritici* Suppressing the Pathogenic-Associated Molecular Pattern-Triggered Immune Response Were Screened by Transient Expression of Wheat Protoplasts

**DOI:** 10.3390/ijms22094985

**Published:** 2021-05-07

**Authors:** Yongying Su, Yanger Chen, Jing Chen, Zijin Zhang, Jinya Guo, Yi Cai, Chaoyang Zhu, Zhongyuan Li, Huaiyu Zhang

**Affiliations:** 1College of Life Science, Sichuan Agricultural University, Ya’an 625014, China; yongysu@163.com (Y.S.); anty9826@163.com (Y.C.); jyguo@sicau.edu.cn (J.G.); yicai@sicau.edu.cn (Y.C.); cygzhu@gmail.com (C.Z.); lizhongyuan12321@163.com (Z.L.); 2Chengdu Institute of Biology, Chinese Academy of Sciences, Chengdu 610041, China; chenjing@cib.ac.cn (J.C.); zhangzj1@cib.ac.cn (Z.Z.)

**Keywords:** wheat stripe rust, transient expression of wheat protoplasts, haustorial effector protein, pathogenic-associated molecular pattern-triggered immune (PTI), pathogenic mechanism

## Abstract

*Puccinia striiformis* f. sp. *tritici* (*Pst*) is an important pathogen of wheat (*Triticum aestivum* L.) stripe rust, and the effector protein secreted by haustoria is a very important component involved in the pathogenic process. Although the candidate effector proteins secreted by *Pst* haustoria have been predicted to be abundant, few have been functionally validated. Our study confirmed that chitin and flg22 could be used as elicitors of the pathogenic-associated molecular pattern-triggered immune (PTI) reaction in wheat leaves and that *TaPr-1-14* could be used as a marker gene to detect the PTI reaction. In addition, the experimental results were consistent in wheat protoplasts. A rapid and efficient method for screening and identifying the effector proteins of *Pst* was established by using the wheat protoplast transient expression system. Thirty-nine *Pst* haustorial effector genes were successfully cloned and screened for expression in the protoplast. We identified three haustorial effector proteins, PSEC2, PSEC17, and PSEC45, that may inhibit the response of wheat to PTI. These proteins are localized in the somatic cytoplasm and nucleus of wheat protoplasts and are highly expressed during the infection and parasitism of wheat.

## 1. Introduction

Wheat stripe rust is a fungal disease caused by *Pst*. Although efforts have been made to control the disease through the use of fungicides, the popularization of disease-resistant wheat cultivars, and the adjustment of seeding times to coincide with host-free periods, stripe rust is still a serious threat to global wheat production [1,2]. The reason is that the species that cause stripe rust undergo rapid mutation and have a strong spreading ability; consequently, cultivars that were previously resistant to stripe rust lose their resistance [3,4]. For example, wheat varieties with the Yr26 resistance gene may be resistant to stripe rust disease dominated by CYR32 and CYR33 in China, but when the physiological varieties V26/CM42 and V26/GUI22 appear, related varieties lose their resistance [1,5].

Plants have evolved specific defense responses to different biological stresses. Pattern recognition receptors (PRRs) on plant cell membranes detect conserved pathogenic-associated molecular patterns (PAMPs) or microbial-associated molecular patterns (MAMPs) that can trigger immune responses, also known as the PAMP-triggered immune (PTI) response. Fungal PAMPs include xylanases and cell wall derivatives such as chitin [6], and bacterial PAMPs include lipopolysaccharides, flagellin (core sequence flg22), and elongation factor Tu (core sequence elf18) [7]. PTI is often accompanied by the accumulation of defense-related proteins, leading to the deposition of callose and the production of antibacterial compounds, such as the burst of reactive oxygen species (ROS), e.g., hydrogen peroxide (H_2_O_2_) [8,9]. Plant PTI successfully resists most pathogenic microorganisms, but a few pathogenic microorganisms have evolved a corresponding survival strategy. Although microorganisms can infect plants by secreting effectors that inhibit PTI, these effector proteins can be resistance proteins (R Proteins) that enhance defense reactions, namely, effector-triggered immunity (ETI) [10,11]. ETI leads to programmed cell death (PCD), known as a hypersensitive response (HR), which enables plants to produce systematic immune responses [12]. Plants and pathogens are coevolutionary, and host defense systems are a major selective force for eradicating pathogen maladaptive effector pools; host plants survive evolution only if they are selected to recognize and resist invading pathogens [13]. The main immune responses of plant–pathogen interactions are PTI and ETI, and the final result depends on the recognition of pathogens by plants and the activation of corresponding immune responses, as well as the inhibition of the pathogens by the plant immune system, both of which are regulated by the hormone signaling network [14].

Many plant-pathogenic microorganisms disrupt host defense responses by secreting effector factors or otherwise enhance their pathogenicity [14,15]. Bacterial pathogens lead to host-induced disease through special type III protein secretion systems (T3SSs), which inject virulence proteins into host cells to suppress PTI and ETI [16,17,18]. Since the effectors of *Oomycetes* have conserved protein sequences, such as RXLR, HaRxLs, and DEER, many effectors of *Oomycetes* have been identified and verified [19,20]. Some effector proteins of *Pseudomonas syringae* and *Xanthomonas*, which use T3SS encoded by *hrp* gene clusters to transport type III secretory effector (T3SE) proteins into plant host cells to cause disease, have been shown to inhibit PTI or ETI [21,22,23,24]. Most effectors secreted by the Type III protein secretion systems in bacterial pathogens have characteristic patterns. According to these unique characteristics, effectors can be distinguished from other proteins by computer algorithms [25]. Unlike bacteria and oomycetes, the prediction of fungal effectors is difficult, and this problem is currently being solved by combining expression analysis and machine learning. However, the large-scale functional analysis of fungal effectors is still limited due to the lack of conserved sequences [26,27].

Although a great deal of research has been done on the effector proteins of plant bacteria, relatively few effector proteins have been verified compared to bacteria because *Pst* is a strictly obligate biotrophic fungus. Similarly to many obligate biotrophic fungi, *Pst* has a special infectious architecture, the haustorium, which is able to penetrate the cell wall and make close contact with the host cell to absorb nutrients from the host cytoplasm [28]. The haustorium has a unique metabolic function, which can produce and deliver effectors to the host cytoplasm, and the expression level of many effector genes is high in the haustorium [29]. Through the use of *Tumefaciens* and bacterial T3SS to deliver candidate effector proteins to the host for transient expression, as well as host-induced gene silencing (HIGS)-mediated *Pst* RNA silencing, the functions of *Pst* virulence effectors, such as PEC6, PSTha5a23, and PST_8713, have been verified [30,31,32,33]. High-throughput sequencing was applied to the genome and haustorial transcriptome of *Pst*, and thousands of highly expressed genes for proteins secreted by the haustorium were discovered by gene expression and polymorphism analysis [34,35,36]. Although the effector protein genes secreted by haustorium are abundant, few of them have been verified. Therefore, it is of great significance to establish a rapid and efficient method of analyzing these genes to understand the pathogenic mechanism of *Pst* and for the control of wheat stripe rust.

A transient gene expression system, using the mesophyll protoplast of *Arabidopsis thaliana*, has proven to be an important and versatile tool that can be used in cell-based experiments to analyze the functions of various signaling pathways and cellular mechanisms by molecular, cellular, biochemical, genetic, genomic, and proteomics methods [37]. In the study of the interaction between bacterial pathogens and plant hosts, it has become a routine method to explore the effects of bacterial effectors in host cells by transient expression of bacterial effectors in host protoplasts [38,39,40]. However, studies on the interaction between wheat and *Pst* using a protoplast transient expression system have not yet been reported. Therefore, based on the results of *Pst* genome resequencing and haustorial transcriptomics, we selected candidate effectors using a bioinformatics method and analyzed their ability to inhibit PTI through the transient expression of wheat protoplasts. In addition, we screened out three effectors that may suppress the PTI response.

## 2. Results

### 2.1. Identification of PTI Reaction Elicitors in Wheat

In the plant PTI signaling pathway, PRRs on the plasma membrane of cells are usually activated by the recognition of corresponding PAMPs to produce ROS. In order to identify PAMPs that stimulate PTI responses in wheat, the accumulation of H_2_O_2_ in leaves was measured by 3,3-diaminobenzidine (DAB) staining. We selected five widely reported plant PTI elicitors, elf18, flg22, chitosan, chitin, and (GlcNAC)_6_, to treat wheat leaves and determined the level of accumulation of H_2_O_2_ in these leaves. The results showed that wheat leaves treated with flg22 showed obvious yellow-brown areas compared with those of the control wheat. Leaves treated with chitosan, chitin, and (GlcNAC)_6_ exhibited an obvious brown color. There was no significant change in the leaves treated with elf18. The results showed that chitosan, chitin, (GlcNAC)_6_, and flg22 could induce H_2_O_2_ burst in wheat leaves, but the ROS level induced by chitosan, chitin, and (GlcNAC)_6_ was higher than that induced by flg22 in wheat leaves (Figure 1).

These results indicated that chitosan, chitin, and (GlcNAC)_6_ significantly elicited a PTI response in wheat leaves, and more than flg22. Chitin is an important component of fungal cell walls. Chitosan and (GlcNAC)_6_ are the products of chitin degradation. Flg22 is the core peptide of bacterial flagellin. Studies in rice have shown that chitin, chitosan, and (GlcNAC)_6_ can stimulate PTI immune signaling, which is consistent with our research results [41]. For this reason, chitin was used as the elicitor of the PTI reaction in wheat in the subsequent research.

### 2.2. Chitin Induces PTI Marker Genes in Wheat

In order to better detect the wheat PTI response, it was necessary to screen marker genes involved in the wheat PTI response. Therefore, we selected 20 genes enriched in the chitin-induced wheat leaf transcriptome and with significant differential expression in the plant–pathogen interaction pathway as marker gene candidates for the PTI response (Appendix A). The gene expression of wheat leaves 4 h post treatment (hpt) was analyzed by quantitative real-time PCR (qRT-PCR), and the relative expression of each gene after chitin treatment was calculated with H_2_O as a control. The expression levels of *TRAESCS4D02G229900*, *TRAESCS3A02G347500*, *TRAESCS5A02G447200*, and *TRAESCS7A02G198800* were at the same level in chitin treated wheat leaves, and the highest relative expression level was *TRAESCS4D02G229900*, at 35.5 times the base level. Student’s *t*-test showed that the relative expression levels of these four genes were significantly higher than those of other genes (*p* < 0.01, Figure 2).

Gene annotation analysis showed that *TRAESCS4D02G229900*, *TRAESCS3A02G347500*, *TRAESCS5A02G447200*, and *TRAESCS7A02G198800* were *TACMPG1-like*, *TAWRKY27*, *TAPDR2-like*, and *TAPR-1-14*. *TaCMPG1-like* is a ubiquitin ligase gene, which is an early immune response gene induced by fungi in *Arabidopsis thaliana* [42,43]. *TaWRKY27* is a homologous gene in the WRKY transcription factor family, which regulates plant growth and development, as well as signal transduction under biotic and abiotic stresses. In *Arabidopsis thaliana*, 49 WRKY transcription factors were found to be involved in the regulation of the immune response during the infection of *Pseudomonas syringae* [44]. The results showed that *TAPDR2-like* could be induced by *Magnaporthe* in wheat [45]. *TaPr-1-14* is a member of the wheat PR1 gene family, which is involved in the ability of the plant cell wall to prevent infection. More than 20 Pr1 genes have been found in wheat [46].

In order to further study the expression characteristics of these four candidate marker genes, their relative expression levels in wheat leaves treated with chitin for 0, 0.5, 1, 2, 4, 8, and 12 h were measured using 0-h samples as the control. As shown in Figure 3, the expression levels of the four candidate marker genes began to increase significantly after 0.5 h of chitin treatment (*p* < 0.05), and their transcription levels reached a highly significant peak at 2–4 hpt (*p* < 0.01) and then showed a decreasing trend. The transcriptional levels of *TaWRKY27* and *TaPr-1-14* reached their highest at 2 hpt, with relative expression levels of 19.6 and 29.5, respectively. The expression levels of *TaCMPG1-like* and *TaPDR2-like* reached extremely significant peaks at 4 hpt (*p* < 0.01), with relative expression levels of 37.5 and 28.5-fold, respectively. In order to establish an efficient effector screening method, in addition to requiring a high expression of PTI marker genes, we also required that PTI marker genes were located downstream of signal transduction. Therefore, we determined that *TaCMPG1-like* and *TaPr-1-14* would be the marker genes for detecting the PTI response in the subsequent analyses.

### 2.3. Transcription Characteristics of Marker Genes of the PTI Response in Wheat Protoplasts Induced by Chitin

The aim of this study was to test whether marker genes could be expressed normally in wheat protoplasts. The relative expression levels of *TacMPG1-like* and *TaPr-1-14* in wheat protoplasts after 1, 2, 4, 6, and 8 h of chitin treatment were measured using 0-h samples as the calculation standard (Figure 4). The results showed that in wheat protoplasts, the expression of *TacMPG1-like* induced by chitin was first upregulated and then decreased after peaking at 4 hpt, with a relative expression level of 36.6 times. The expression of *TacMPG1-like* at 2, 4, and 6 hpt levels was at the same level and was significantly higher than those at other time points. The expression of *TaPr-1-14* was upregulated and then downregulated, with a highly significant peak at 2 hpt (29.5 times the relative expression level). The expression levels of *TaPr-1-14* at 4 hpt and 2 hpt were similar. The results showed that the two marker genes had the same tendency of being induced by chitin in protoplasts and leaves, so these two genes could be used as marker genes of the PTI response in wheat protoplasts.

### 2.4. Verification of Wheat Protoplast Transient Expression System for Screening Effectors

Pathogenic effector PSTha5a23 inhibited the immune responses induced by Bax, PAMP-INF1, MKK1, and NPK1 in *N. benthamiana*, and it also suppressed the PTI response in wheat leaves [30]. In order to study the effect of pathogenic effectors on the signal transduction pathway of PTI response in wheat protoplasts, in this experiment, the expression vectors pHBT95-35S-PSTha5a23-HAHA and pHBT95-35S-GFP were expressed in wheat protoplasts for 8 h and were then treated with H_2_O and chitin for 4 h. The gene expression level of *TaPr-1-14* was analyzed by qRT-PCR using the H_2_O treatment as the calculation standard. As shown in Figure 5A, the relative expression level of the marker gene in the chitin-treated green fluorescent protein (GFP)-expressing wheat protoplast was 36.6 times, while the relative expression level of the marker gene expressing PSTha5a23-HAHA was only 17.2. The results showed that the pathogenic factor PST ha5a23 could inhibit the expression of *TaPr-1-14*, a chitin-induced marker gene, in wheat protoplasts. Both PSTha5a23-HAHA and GFP proteins can be expressed in wheat protoplasm and can be detected by corresponding labeled antibodies (Figure 5B). In addition, the protoplasts expressing GFP in this study had clear fluorescence signals, and the GFP transformation efficiency of cell count statistics exceeded 85%, indicating that the gene had a good transformation and expression effect in protoplasts (Figure 6). Therefore, we believe that this wheat protoplast transient expression system is effective for large-scale screening and functional research of effectors.

### 2.5. Screening of Effectors Suppressing the PTI Response in Wheat

Based on the genomic database of CYR32 and PST79 of stripe rust species and the haustorial transcriptome database of 104E137A of stripe rust species, 328 *Pst* genes encoding haustoria-secreted proteins were obtained through a bioinformatics analysis. In total, 50 genes encoding fewer than 500 amino acids were selected as candidate effector proteins for cloning, and 39 haustorial effector proteins of *Pst* were cloned successfully, with a success rate of 78% (Appendix A).

In wheat protoplast cells expressing the control vector pHBT-35S-GFP, the relative expression level of the marker gene *TaCMPG1-like* was 39.1 times after chitin treatment, using H_2_O-treated samples as a reference (Figure 7A). Among the protoplasts expressing candidate effector proteins, the relative expression levels of *TaCMPG1-like* in six protoplasts expressing effector proteins were significantly decreased after chitin treatment (21.3-, 19.8-, 22.1-, 26.9-, 19.9-, and 25.9-fold, *p* < 0.05); the six effector proteins were PSEC2, PSEC17, PSEC18, PSEC23, PSEC34, and PSEC45. When other effector proteins were expressed in protoplast cells, the relative expression level of TACMPG1-like was not significantly different from that in protoplast cells expressing GFP.

In the protoplasts expressing the candidate effector protein, the transcription level of the *TaPr-1-14* marker gene in only three protoplasts expressing the effector protein was significantly decreased after chitin treatment (19.2-, 17.8-, and 20.9-fold) compared with the protoplasts expressing GFP (34.6-fold, *p* < 0.05) (Figure 7B). These three effectors were PSEC2, PSEC17, and PSEC45, and their corresponding gene IDs are *Pstv_15892*, *Pstv_12697*, and *Pstn_5645*, respectively. Bioinformatics analysis of the candidate effector proteins PSEC2, PSEC17, and PSEC45 revealed that the three proteins encoded 187, 257, and 230 amino acids, respectively. They had no transmembrane region and no conserved protein domain, and the N-terminal signal peptide sequence was 18, 19, and 22, respectively (Appendix A). Considering that *PSEC2*, *PSEC17*, and *PSEC45* all have inhibitory effects on the expression of *TaCMPG1-like* and *TaPr-1-14*, the *PSEC2*, *PSEC17*, and *PSEC45* genes were selected as the next functional verification genes in this study.

### 2.6. Overexpression of PSED2, PSED17, and PSED45 in Wheat Suppress PTI-Related Callose Deposition

Since PSED2, PSED17, and PSED45 suppressed PTI marker gene expression induced by chitin in wheat protoplasts, we decided to further evaluate their ability to inhibit host PTI responses. Effector detector vectors (EDV) pEDV6 can use bacterial TTSS to deliver individual non-bacterial effectors to host plant cells [47]. pEDV6 was delivered into wheat by the modified *P. fluorescens* strain effector to host analyzer (EtHAn), which carries a functional TTSS [48]. EtHAn is non-pathogenic on wheat, and there were no chlorosis or necrosis phenotypes in wheat leaves inoculated with EtHAn (Figure 8A). However, callose deposition was observed on wheat leaves after inoculating with EtHAn (Figure 8B), indicating that infection with EtHAn could induce wheat PTI responses. EtHAn strain-carrying GFP was used as positive control. The pEDV6:GFP, pEDV6:PSED2, pEDV6:PSED17, and pEDV6:PSED45 constructs were transfected into EtHAn strains, which were infiltrated into wheat leaves. Callose deposition was observed in the leaves of wheat inoculated with EtHAn strains carrying pEDV6:GFP, pEDV6:PSED2, pEDV6:PSED17, or pEDV6:PSED45, but callose deposition was significantly reduced in the pEDV6:PSED2-, pEDV6:PSED17-, and pEDV6:PSED45-inoculated wheat leaves compared with the pEDV6:GFP-inoculated wheat leaves (Figure 8B). The callose deposition was measured in the infiltrated wheat leaves and the callose deposition of PSED2-, PSED17-, and PSED45-infiltrated leaves significantly decreased by 58.7%, 52.1%, and 45.6% compared to wheat leaves introduced with the EtHAn strain carrying pEDV6:GFP (Figure 8C). These results indicate that overexpression of PSED2, PSED17, and PSED45 in wheat suppress PTI-related callose deposition.

### 2.7. The Candidate Effectors PSEC2, PSEC17, and PSEC45 Are Highly Induced In Planta

In the process of *Pst* infection of wheat, uredospore germination occurs first, after the germination tube that forms over the stomatal aperture enters the leaf interior through the stoma, where it differentiates into a substomatal vesicle. Then, infection hyphae begin to grow. They form haustorial mother cells outside the cell wall and finally break through the cell wall to form haustorium, and the infection usually takes about 24 h to complete [29]. The results showed that, compared with the ungerminated uredospore, the relative gene expression of *PSEC2* peaked at 24 h post infection (hpi) and was consistently high (18–210 hpi) during the infection stage of *Pst* mycelial differentiation (*p* < 0.05). *PSEC17* and *PSEC45* were greatly induced and expressed during *Pst* infection of the host. The first peak of gene relative expression occurred at 24–48 hpi, at the beginning of the biotrophic parasitic stage, and the second peaks occurred at 158–210 hpi and 48–72 hpi in the later stages of biotrophic parasitism and spore production, respectively (Figure 9). These results indicate that these three effectors may suppress the immune signal transfection of the host after stripe rust infects wheat leaves and promote the proliferation of stripe rust in the wheat leaves.

### 2.8. PSEC12, PSEC17, and PSEC45 Are Localized to the Wheat Cytoplasm and Nucleus

After being secreted from pathogens, the effector protein enters the host cell and can be transferred to different organelles [49]. To determine the subcellular localization of PSEC2, PSEC17, and PSEC45, transient expression analyses of their GFP fusion protein in wheat mesophyll protoplasts were conducted. In control samples, GFP was expressed by the 35S promoter of the cauliflower mosaic virus, and the fluorescent signal was transmitted throughout the whole cell (cytoplasm and nucleus). The fluorescence of Psec2, Psec17, and Psec45 fusion proteins in the cytoplasm and nucleus of wheat mesophyll protoplasts was also observed (Figure 10). These results indicated that the protein expressions of Psec2-GFP, Psec17-GFP, and Psec45-GFP were localized in the cytoplasm and nucleus of wheat.

## 3. Discussion

Pathogenic effectors are important components that suppress the host immune system. Many plant pathogens destroy the host immune system by releasing effector proteins, making it more susceptible to disease [50]. Studies have reported the expression of proteins in plants by the bacteria T3SS and *Tumefaciens*, as well as the analysis of effector proteins by methods such as HIGS, but there is no routine high-throughput effector protein screening method [51,52,53]. In this study, a protoplast transient expression system related to the wheat PTI response was established by identifying the elicitors and marker genes. The system could detect the pathogenic effectors of stripe rust fungus with high throughput, three possible effectors that might suppress the wheat PTI response were identified, and the mechanism of action was preliminarily discussed.

Although there have been many studies on PAMPs in other plants, there are few reports on the response of pathogenic PAMPs in triggering early PTI in wheat [54]. Histological observation of treated wheat leaves by DAB staining showed that Flg22, chitosan, chitin, and (GlcNAC)_6_ could induce a PTI response in wheat leaves. As chitin is the main component of the cell wall of pathogenic *Pst*, chitin was selected as the elicitor to induce the PTI response in wheat in this study.

When plant leaves are treated with chitin, extracellular signals can be transferred into the cell through the recognition of receptors on the cell membrane, and the signal is transfected to the MAPK pathway. MAPK-related proteins are phosphorylated and activate transcription factors in the nucleus to regulate the expression of downstream defense-related proteins, such as PR protein [55]. Although the biological function of PR-1 remains obscure, there is substantial evidence that PR-1 is involved in cellular defense during pathogen interactions with plants [56]. In order to better detect the PTI response in wheat, downstream genes related to immune signal transduction with higher expression levels should be selected as marker genes of the PTI response. In total, 20 differentially expressed genes were detected by qRT-PCR, and four genes were significantly upregulated in the chitin-induced PTI response in wheat leaves. The four genes were *TaCMPG1-like*, *TaWRKY27*, *TaPDR2-like*, and *TaPr-1-14*, and their expression in wheat mesophyll protoplasts was consistent with the experimental results. On this basis, *TaPr-1-14* was selected as the main marker gene for the PTI response in wheat, providing a new method for the detection of the PTI response in wheat.

The mesophyll protoplast transient expression system is an important and convenient tool for exploring the signal transduction of the interaction between model plants and pathogens [37]. *FRK1* is a marker gene for the PTI response induced by flg22 in *Arabidopsis thaliana* [57]. Detection of flg22-induced *FRK1* gene expression by the expression of candidate effector proteins in *Arabidopsis* protoplasts is a popular method used to analyze the pathogenicity of bacterial effector proteins [58,59,60]. However, in the process of analyzing the pathogenic mechanism of interaction between *Pst* and wheat, no relevant reports were found on the study of fungal effectors using the transient transformation system of wheat mesophyll protoplasts. We expressed the reported virulence effector PSTha5a23 in wheat protoplasts, and the results showed that it could suppress the expression of the chitin-induced marker gene *TaPr-1-14*, which indicated that it was feasible to identify virulence effectors of stripe rust by using the wheat protoplast transient expression system.

Due to the development of genome sequencing technology, various physiological species of *Pst* have been re-sequenced, and a large number of haustorial secretion protein genes have been found [2]. Currently, the methods used to analyze whether these genes are effectors rely mainly on HIGS and the *Pseudomonas fluorescens* strain EtHAn [53]. However, both methods require the infection of wheat leaf tissues and observation of results after several weeks, especially the in vitro transcriptase treatment of viral vectors by HIGS, which is also expensive. Therefore, we established a screening method for effector proteins in wheat mesophyll protoplasts: the candidate effectors of stripe rust were expressed in protoplasts, and the protoplasts were treated with chitin to detect whether the expression of PTI marker genes was suppressed, to determine whether the candidate proteins were pathogenic. Compared with HIGS and EtHAn, this method has a shorter detection cycle, simpler operation, and high throughput screening for multiple candidate effector proteins, which provides a more effective means for the study of the interaction between *Pst* and wheat.

During the interaction between pathogenic bacteria and plants, pathogenic microorganisms secrete and transport pathogenic effector proteins into host cells to promote their infection of the host and their own growth and reproduction. Prokaryotic pathogens release a variety of effectors in host cells through T3SS and inhibit host immune defenses by affecting the molecular, biochemical, and physiological functions of host cells [61,62,63]. Unlike bacteria, stripe rust does not have T3SS and forms haustoria in the host cell wall during infection. The secreted proteins of haustorial cells are transported out of the cell and released into wheat cells by 15–30 hydrophobic signal peptides at the N-terminal [64,65]. Existing haustorial transcriptome studies indicated that *Pst* has about 600 haustorial secretion proteins [28,66].

Bioinformatics analysis was used to evaluate and predict the secreted proteins of haustoria. In total, 39 candidate effectors were expressed in wheat protoplasts, among which, three effectors, i.e., PSEC2, PSEC17, and PSEC45, could suppress the expression of PTI marker genes. In this study, we demonstrated that overexpression of PSED2, PSED17, and PSED45 in wheat suppressed PTI-related callose deposition; these results indicate that PSED2, PSED17, and PSED45 can inhibit wheat PTI responses. Our results showed that these three effectors were all localized in the cytoplasm and nucleus of wheat mesophyll protoplasts, consistent with the reported localization results of stripe rust pathogenic effectors PEC6 and PST_8713, and we deduced that these three effectors may play a role in the host cytoplasm or nucleus. The gene expression of the effectors showed that *PSEC2* was highly expressed about 6 h earlier than *PSEC17* and *PSEC45*, suggesting that *PSEC2* may have been involved in the suppression of immune defense during the growth and impregnation of mycelia. *PSEC17* and *PSEC5* may have high transcriptional levels after the formation of haustoria in host cells, which participate in the suppression of wheat PTI immune signal transference and enhance the ability of stripe rust to infect wheat. The gene expression levels of effectors were different during host infection and parasitism, which might be due to the different suppression effects of effectors on immune defense at different infection times.

Clarifying the function of genes associated with the pathogenicity of *Pst* and its impact on host targets should help us to identify the components involved in host defense, which could help breeders select genes that are more resistant against *Pst*. Therefore, the specific mechanisms through which PSEC2, PSEC17, and PSEC45 affect plant defense and rust pathogenicity will be further studied in the future to provide a basis for the development of wheat stripe rust control strategies.

## 4. Materials and Methods

### 4.1. Plant Materials and Inoculation

Two wheat cultivars, Chuannong 19 (CN19) and Mianyang 11 (MY11), and stripe rust *Pst* pathotype CYR32 were used in this study. CN19 possesses the stripe rust resistance gene Yr41, which has resistance to CYR32 at the adult plant stage [67], while Mianyang11 is susceptible to CYR32. At the seedling stage, CN19 has wide leaves, which are suitable for the preparation of mesophyll protoplasm cells, and it was cultured under the conditions of a short photoperiod (12 h light at 23 °C/12 h darkness at 20 °C) for later use. In order to study the gene expression characteristics of the candidate effectors of stripe rust during infection, MY11 was grown to the jointing stage under normal light conditions, and wheat leaves were inoculated with stripe rust. Wheat leaves were collected at 12, 18, 24, 36, 48, 72, 115, 158, and 210 h post-CYR32 inoculation to extract RNA for qRT-PCR. Each treatment was performed with three independent biological replicates.

### 4.2. Detection of the H_2_O_2_ Accumulation Level

In this study, ddH_2_O, 1 μM elf18 (Sangon, Shanghai, China), 1 μM flg22 (Sangon, Shanghai, China), 100 μg/mL Chitosan (Sigma, Beijing, China), 100 μg/mL Chitin (Sigma, Beijing, China), and 100 μg/mL (GlcNAC)_6_ (Qingdao BZ Oligo Biotech, Qingdao, China) were injected into wheat CN19 seedling leaves, and each treatment group had 3 biological replicates per sample. After 6 h of treatment, the treated wheat leaves were stained with 5 mM DAB dissolved in 10 mM 4-Morpholineethanesulfonic acid (MES) (pH 3.8) for 8 h in darkness. Then, the stained samples were decolorized in glycerol: acetic acid: ethanol (1:1:3, *v*/*v*/*v*) in a boiling water bath for 0.5–2 h and photographed, as described previously [68].

### 4.3. Total RNA Extraction and Transcription Level Analysis

Leaf or mesophyll protoplast samples were collected, and total RNA was extracted with Trizol reagent (Takara, Dalian, China) according to the manufacturer’s instructions. DNase I was used to remove contaminated genomic DNA from the samples. First-strand cDNA was synthesized using a reverse transcriptase kit (Takara, Dalian, China), according to the manufacturer’s instructions. qRT-PCR was performed, and SYBR Green I Master Mix (Takara, Dalian, China) was used in a volume of 25 μL with the primers in a CFX96 RT-PCR system. Wheat *TaActin* was used as the reference gene for marker gene expression analysis, and stripe rust *EF1* was used as the reference gene for effector transcription level analysis. The corresponding primers are shown in the supporting information (Appendix A). All qRT-PCR reactions were repeated three times. The relative expression of each gene was calculated using the 2^−ΔΔCT^ method [69].

### 4.4. Effector Selection and Plasmid Construction

Based on the transcriptome sequencing of haustoria and genome resequencing of stripe rust [28,35,66], the following bioinformatic methods were used for prediction: SignalP 4.1 (http://www.cbs.dtu.dk/services/SignalP, accessed on 10 May 2020), an online protein signal peptide prediction software package, was used to predict the signal peptide, and proteins with the signal peptide were selected, with a size less than 500 amino acids; the conserved domain of the protein was analyzed by PFAM (http://pfam.xfam.org/, accessed on 10 May 2020); and the protein without Pfam motif was selected.

The primers used for plasmid construction are shown in the supporting information (Appendix A). With the use of cDNA from CN19 leaf tissues inoculated with CYR32 as a template, the open reading frames (ORFs) of 39 candidate effector genes were amplified by FastPfu DNA polymerase (Novoprotein, Shanghai, China), which did not contain signal peptide sequences. The effector fragments of PCR amplification were introduced into the pEDV6, pHBT95-35S-HA, or GFP vector (Accession: KX510275) [70] by NovoRec^®^ Plus Recombinase (Novoprotein, Shanghai, China) for transient expression in wheat protoplasts.

### 4.5. Transient Expression and Analysis in Protoplasts

Mesophyll protoplasts were prepared from CN19 leaves cultured for 14–20 days. Wheat mesophyll protoplast isolation, PEG-calcium transfection of plasmid DNA, and protoplast culturing were performed by referring to the preparation methods for *Arabidopsis thaliana* and wheat mesophyll protoplasts [37,71]. To detect the expression level of marker genes induced by chitin in mesophyll protoplasts, wheat protoplasts (200 μL) were added to 2 μL 10 mg/mL chitin at 23 °C and exposed to 2000 lux for 1, 2, 4, 6, and 8 h. The cells were collected under 200 G centrifugal force and immediately frozen in liquid nitrogen and stored at −80 °C for marker gene transcription analysis.

In order to screen the pathogenic effector inhibiting the wheat PTI reaction, the expression vectors of candidate effectors were transformed into the prepared wheat protoplasts, and the protoplasts of the transformed pHBT95-35S-GFP vector were used as the negative control. After 8–10-h culture, the protoplasts were treated with H2O and chitin for 4 h. H_2_O treatment was used as the calculation standard, and the relative expression levels of marker genes *TaCMPG1-like* and *TaPr-1-14* were determined by qRT-PCR.

In order to determine the subcellular localization of PSEC2, PSEC17, and PSEC45 in wheat mesophyll protoplasts, the GFP fusion vectors PSEC2-GFP, PSEC17-GFP, and PSEC45-GFP were constructed after their signal peptide sequences were removed and transformed into protoplasm for expression. The next day, their images were observed and captured under the Olympus FV1200 confocal microscope (Olympus, Tokyo, Japan).

### 4.6. Western Blotting Analysis

To determine protein expression in wheat protoplasts, Western blotting was performed as previously described [30]. The total proteins in wheat mesophyll protoplasts were extracted by the method as described [68], and the extracted protein solution was separated by 12% SDS-PAGE gel. The proteins were transferred to a microporous polyvinylidene fluoride (PVDF) membrane and incubated in the blotting buffer (5% non-fat milk powder in TBS). Proteins were detected using mouse-derived GFP or HA antibodies (CWBio, Beijing, China) incubated overnight at 4 °C. The membranes were incubated with horseradish peroxidase-conjugated anti-mouse secondary antibody (CWBio, Beijing, China) and developed using a chemiluminescent detection system (Bio-Rad, Hercules, CA, USA).

### 4.7. Bacterial T3SS-Mediated Overexpression in Wheat Leaves

The pEDV6:GFP (control), pEDV6:PSED2, pEDV6:PSED17, and pEDV6:PSED45 constructs were transformed into P. fluorescens strain EtHAn by electroporation. For the infiltration of leaves, recombinant strains of EtHAn were grown in King’s B liquid medium for 48 h at 28 °C, harvested, and resuspended in an infiltration medium (10 mM MgCl_2_). EtHAn suspensions were infiltrated at an OD600 of 0.8 into the third leaves of wheat cultivars (MY11) at the two-leaf stage using a syringe without a needle. The infiltrated wheat plants were maintained in a growth chamber at 25 °C for 2 days. To check the suppression of callose deposition, the pEDV6:GFP (control), pEDV6:PSED2, pEDV6:PSED17, and pEDV6:PSED45 inoculated wheat (MY11) plants were stained with aniline blue, as described previously [72]. The wild type strain EtHAn was used as a PTI trigger in wheat (MY11). Wheat leaves infiltrated with recombinant EtHAn carrying the GFP were used as control.

## Figures and Tables

**Figure 1 ijms-22-04985-f001:**
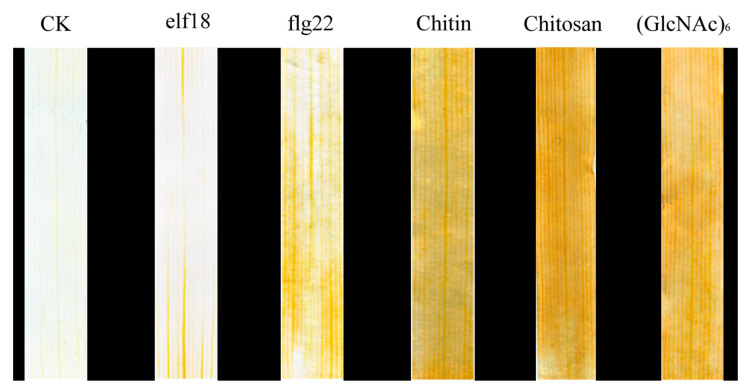
Reactive oxygen species (ROS) induced by pathogenic-associated molecular patterns (PAMPs) in the leaves of wheat cultivar CN19. Histochemical analysis for hydrogen peroxide (H_2_O_2_) by 3,3-diaminobenzidine (DAB) staining. The DAB staining was determined after wheat were treated with 1 μM elf18, 1 μM flg22, 100 μg/mL Chitosan, 100 μg/mL Chitin, or 100 μg/mL (GlcNAC)_6_ for 6 h. CK, untreated wheat plants. The deeper the change in leaf color, the more H_2_O_2_ is accumulated, as indicated by a test from three independent biological replicates.

**Figure 2 ijms-22-04985-f002:**
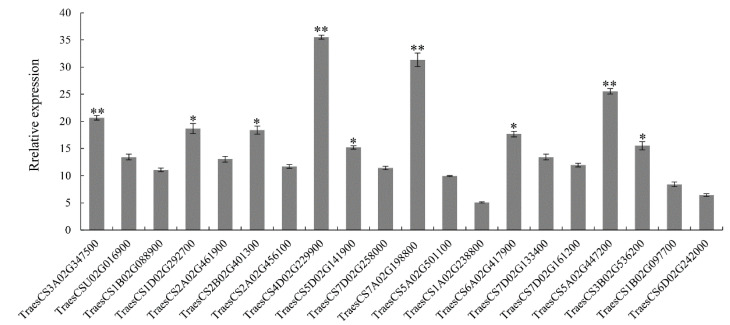
Relative expression analysis of 20 candidate marker genes in the leaves of wheat cultivar CN19 treated with chitin for 4 h. The relative expression of 20 candidate marker genes were examined by qRT-PCR, and the expression levels of these genes were calculated by using the comparative threshold (2^−ΔΔCT^) method, and values are expressed relative to an endogenous wheat reference gene *TaActin*. Bars represent standard deviations (SD), which were calculated from 3 independent biological replicates (*n* = 3). Asterisks indicate significant differences calculated using Student’s *t*-test. (* *p* < 0.05, ** *p* < 0.01).

**Figure 3 ijms-22-04985-f003:**
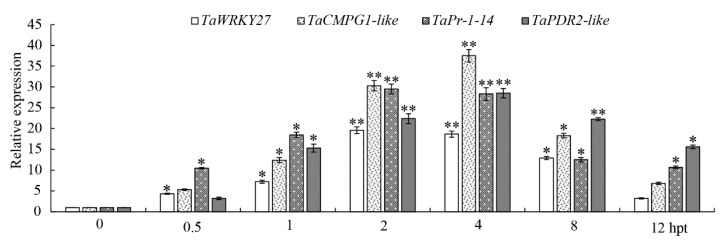
Expression characteristics of four pathogenic-associated molecular pattern-triggered immune (PTI) candidate marker genes induced by chitin in the leaves of wheat cultivar CN19. The relative expression of the four genes was examined by qRT-PCR. Bars represent standard deviations (SD), which were calculated from 3 independent biological replicates (*n* = 3). Asterisks indicate significant differences calculated using Student’s *t*-test. (* *p* < 0.05, ** *p* < 0.01). 0 hpt, un-treated wheat plants. 0.5–12 hpt represent 0.5, 1, 2, 4, 8, and 120 h post-treatment.

**Figure 4 ijms-22-04985-f004:**
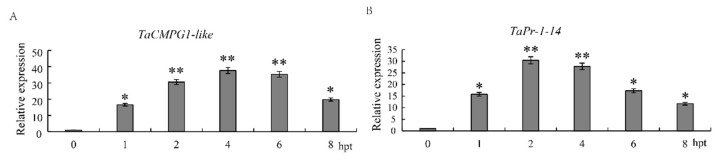
Expression characteristics of marker genes *TacMPG1-like* (**A**) and *TaPr-1-14* (**B**) induced by chitin in wheat protoplasts. The relative expression of t *TacMPG1-like* and *TaPr-1-14* was calculated by using the comparative threshold (2^−ΔΔCT^) method, and values are expressed relative to an endogenous wheat reference gene *TaActin*. Bars represent standard deviations (SD), which were calculated from 3 independent biological replicates (*n* = 3). Asterisks indicate significant differences calculated using Student’s *t*-test. (* *p* < 0.05, ** *p* < 0.01). Wheat protoplasts were isolated from wheat cultivar CN19. 0 hpt, un-treated wheat protoplasts. 1–8 hpt represent 1, 2, 4, 6, and 8 h post-treatment.

**Figure 5 ijms-22-04985-f005:**
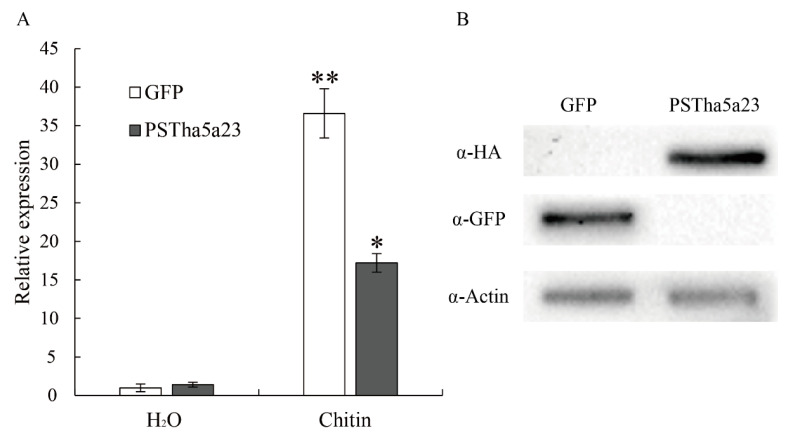
Expression of PSTHa5a23 in wheat inhibits PAMP-triggered immunity (PTI). (**A**) Transient expression of PSTHa5a23 inhibits chitin-induced *TaPr-1-14* expression in wheat protoplasts. The wheat protoplasts isolated from wheat cultivar CN19 were transfected with pHBT95-35S-GFP (GFP) or pHBT95-35S-PSTha5a23-HAHA (PSTha5a23) vector, induced with 100 μg/mL Chitin for 4 h. The relative expression of *TaPr-1-14* was determined by using the comparative threshold ((2^−ΔΔCT^) method, and values are expressed relative to an endogenous wheat reference gene *TaActin*. Values were normalized to an internal H_2_O/GFP control. (**B**) Immunoblot analyses of PSTha5a23-HAHA and GFP proteins obtained from the wheat protoplasts. Each data point represents four replicates. Error bars indicate SD. Student’s *t*-test was carried out to determine the significance. *, *p* < 0.05; **, *p* < 0.01.

**Figure 6 ijms-22-04985-f006:**
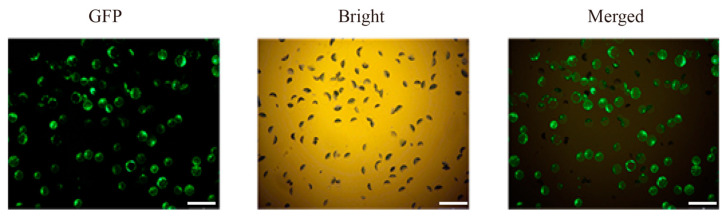
Protoplast of wheat transformed from green fluorescent protein (GFP). Bar = 100 µm.

**Figure 7 ijms-22-04985-f007:**
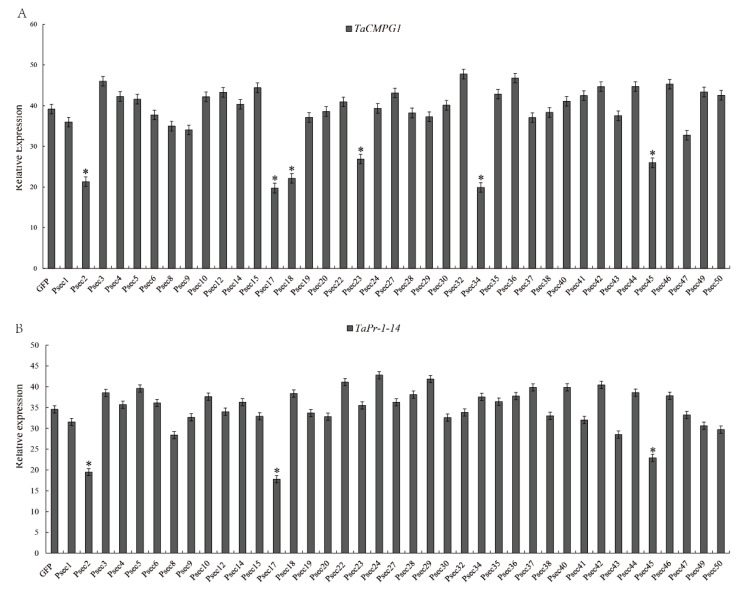
Screening of candidate effectors of stripe rust by the marker gene *TaCMPG1* (**A**) and *TaPr-1-14* (**B**). Wheat protoplasts were isolated from wheat cultivar CN19. Expressing GFP and candidate effectors were treated with chitin for 4h. The relative expression of *TaCMPG1* and *TaPr-1-14* was determined by qRT-PCR. The induction fold of *TaCMPG1* and *TaPr-1-14* was calculated by the gene expression level in chitin-treated protoplasts relative to that in H_2_O-treated protoplasts at the same time point. Bars represent standard deviations (SD), which were calculated from four independent biological replicates (*n* = 4). Asterisks indicate significant differences calculated using Student’s *t*-test. (* *p* < 0.05).

**Figure 8 ijms-22-04985-f008:**
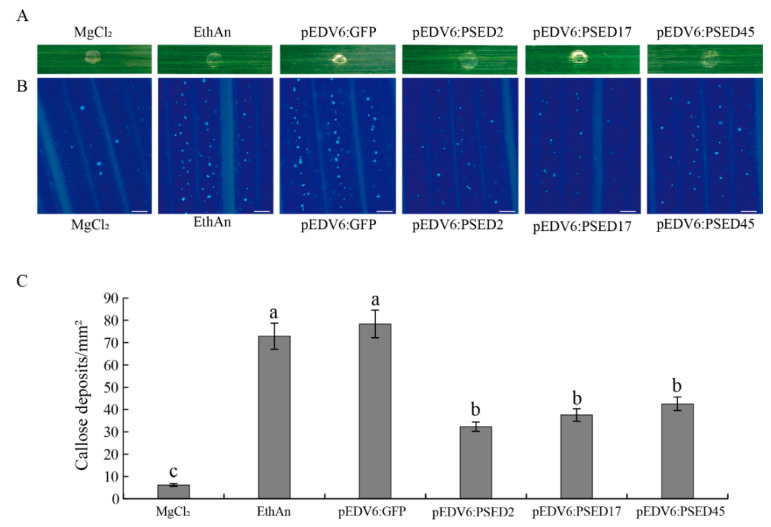
Overexpression of PSED2, PSED17, and PSED45 in wheat (MY11) suppressed PTI-related callose deposition. (**A**) Phenotypes of the wheat leaves infiltrated with the MgCl2 buffer (MOCK), EtHAn, pEDV6:GFP, pEDV6:PSED2, pEDV6:PSED17, or pEDV6:PSED45 at 72 h after infiltration. (**B**) The wheat leaves inoculated with MgCl2 buffer, EtHAn, pEDV6:GFP, pEDV6:PSED2, pEDV6:PSED17, or pEDV6:PSED45 were examined for callose deposition by epifluorescence microscopy after aniline blue staining. (**C**) Average number of callose deposits/mm^2^ in wheat leaves inoculated with MgCl_2_ buffer, EtHAn, pEDV6:GFP, pEDV6:PSED2, pEDV6:PSED17, or pEDV6:PSED45. Bars indicate means of five independent biological replicates with 20 unit areas per replicate (±SD). Different letters note significant differences among different treatments (*p* < 0.05) following Duncan’s multiplication range test.

**Figure 9 ijms-22-04985-f009:**
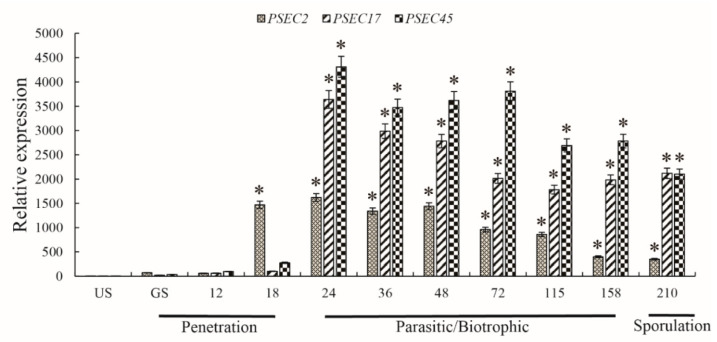
Transcription profiles of *PSEC2 PSEC17*, and *PSEC45* based on quantitative real-time PCR (qRT-PCR). US, urediniospore stage; GS, germ tube stage; 12–210 h, 12–210 h post-inoculation of CYR32 in wheat MY11. The relative gene quantification was calculated by the comparative threshold (2^−ΔΔCT^) method with *Pst* endogenous gene *EF1* as an internal standard and was relative to that of US. Bars indicate means of three independent biological replicates (±SD). * indicates *p* < 0.05.

**Figure 10 ijms-22-04985-f010:**
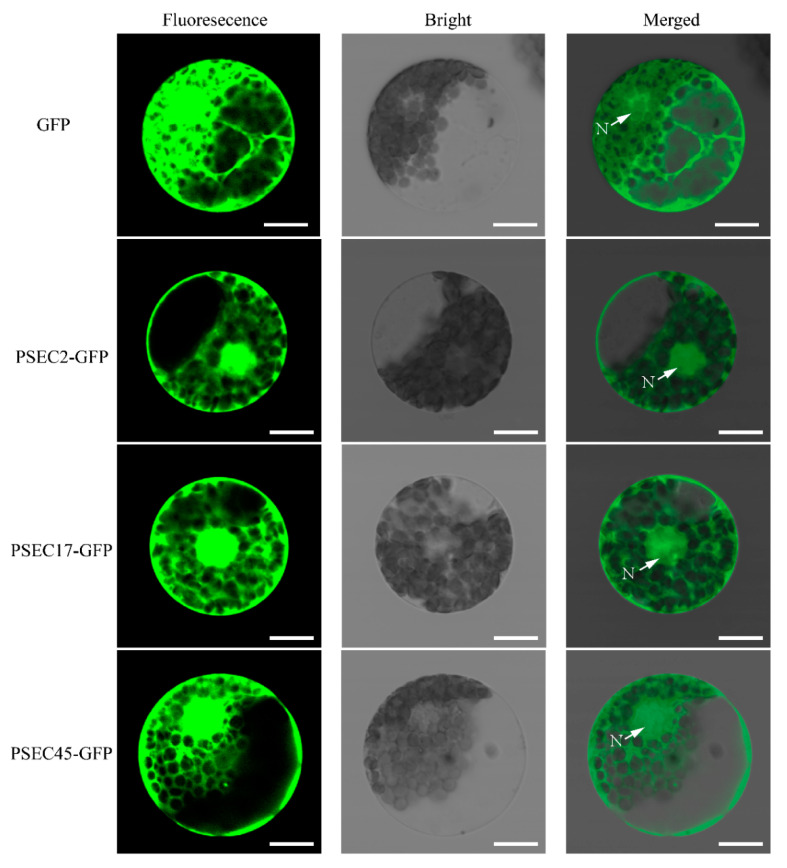
Subcellular localization of PSEC2, PSEC17, and PSEC45 in wheat protoplasts. GFP, PSEC2-GFP, PSEC17-GFP, and PSEC45-GFP fusion proteins were expressed in wheat protoplasts following PEG-mediated transformation. N, nucleus. Bar = 20 µm.

## Data Availability

Not applicable.

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
