# Peer review of "Effectors of Puccinia striiformis f. sp. tritici Suppressing the Pathogenic-Associated Molecular Pattern-Triggered Immune Response Were Screened by Transient Expression of Wheat Protoplasts"

_ijms, 2021, doi:10.3390/ijms22094985_

Round 1

Reviewer 1 Report

A nice manuscript. I advise an accept in present form subject to one minor revision. Line 34 Pst should be stripe rust.

Author Response

We gratefully thank the editor and the reviewer for their time spent making their constructive remarks and useful suggestions, which have significantly raised the quality of the manuscript and enabled us to improve the manuscript. Each suggested revision and comment brought forward by reviewer was accurately incorporated and considered. Below are the comments of the reviewer’s response point-by-point and the revisions are indicated.

Point: Line 34 Pst should be stripe rust.

Response : Thank for your nice suggestion. the statements of "Pst" were corrected as " stripe rust ". (line 34)

Reviewer 2 Report

The authors describe the development of a novel method for screening Pst effector proteins using a wheat protoplast transient expression system. The authors found that TaCMPG1-like and TaPr-1-14 genes can be used as a marker for PTI of Pst infection in the protoplast system. Focusing on these genes expression levels, the authors claimed that three Pst effectors were identified.

Main issues

  1. This manuscript lacks proper data set for characterization of the identified three effector genes (PSEC12, PSEC17 and PSEC45). This weakens the reliability of the screening system. The most critical issue is that there are no data for confirmation of the inhibitory functions to PTI reaction in the identified genes.

Other issues

  1. Why did you remove the signal sequences in the experiment shown in Fig.9? In order to investigate the subcellular localization of proteins, the signal sequences must be essential.
  2. Not only gene ID but also common gene names and (predicted) functions should be explained.
  3. What does "exposed to 2000 1x for" mean? (line 470)
  4. The merged images should contain nuclear staining (fig.9)

Round 2

Reviewer 2 Report

The authors added the data showing an inhibitory effect of the identified genes on PTI. This revision successfully improved the reliability of this manuscript, and the other authors' answers satisfied me. Therefore, I think this manuscript can be published in IJMS.